# A Systematic Review of the Use of Circulating Cell-Free DNA Dynamics to Monitor Response to Treatment in Metastatic Breast Cancer Patients

**DOI:** 10.3390/cancers13081811

**Published:** 2021-04-10

**Authors:** Elisabeth M. Jongbloed, Teoman Deger, Stefan Sleijfer, John W. M. Martens, Agnes Jager, Saskia M. Wilting

**Affiliations:** Department of Medical Oncology, Erasmus University MC Cancer Institute, Dr. Molewaterplein 40, PO Box 2040, 3000 CA Rotterdam, The Netherlands; t.deger@erasmusmc.nl (T.D.); s.sleijfer@erasmusmc.nl (S.S.); j.martens@erasmusmc.nl (J.W.M.M.); a.jager@erasmusmc.nl (A.J.)

**Keywords:** metastatic breast cancer, circulating tumor DNA, liquid biopsies, treatment response

## Abstract

**Simple Summary:**

Currently, the most commonly used method to monitor response to treatment in metastatic breast cancer patients is by radiological imaging. However, these imaging techniques are relatively insensitive and give little to no insight into biological tumor characteristics that might be relevant for the choice of treatment. Circulating tumor DNA (ctDNA), released by tumor cells into the blood of cancer patients, can be used to overcome these shortcomings. Besides the fact that specific alterations are known to predict response to treatment and development of resistance, the total amount of ctDNA is believed to reflect the proliferation rate of the tumor, suggesting ctDNA levels can be used as a general tool to evaluate treatment response. Different methods are available to measure ctDNA primarily based on detection of cancer-specific somatic mutations, DNA methylation, and copy number variations. In this review we have critically analyzed recently published studies using blood-derived ctDNA of metastatic breast cancer patients on multiple time points to monitor disease response in respect to analytical validity and clinical utility.

**Abstract:**

Monitoring treatment response in metastatic breast cancer currently consists mainly of radiological and clinical assessments. These methods have high inter-observer variation, suboptimal sensitivity to determine response to treatment and give little insight into the biological characteristics of the tumor. Assessing circulating tumor DNA (ctDNA) over time could be employed to address these limitations. Several ways to quantify and characterize ctDNA exist, based on somatic mutations, copy number variations, methylation, and global circulating cell-free DNA (cfDNA) fragment sizes and concentrations. These methods are being explored and technically validated, but to date none of these methods are applied clinically. We systematically reviewed the literature on the use of quantitative ctDNA measurements over time to monitor response to systemic therapy in patients with metastatic breast cancer. Cochrane, Embase, PubMed and Google Scholar databases were searched to find studies focusing on the use of cfDNA to longitudinally monitor treatment response in advanced breast cancer patients until October 2020. This resulted in a total of 33 studies which met the inclusion criteria. These studies were heterogeneous in (pre-)processing procedures, applied techniques and design. An association between ctDNA and treatment response was found in most of the included studies, independent of the applied assay. To implement ctDNA-based response monitoring into daily clinical practice for metastatic breast cancer patients, sample (pre-) processing procedures need to be standardized and large prospectively collected sample cohorts with well annotated clinical follow-up are required to establish its clinical validity.

## 1. Introduction

In the last decade, various new therapeutic regimens have been introduced for patients with metastatic breast cancer (MBC). The availability of various treatment regimens urges the development of dedicated tools for adequate monitoring of response to treatment. Currently treatment response in MBC is in general monitored by imaging, which could be evaluated by the Response Evaluation Criteria in Solid Tumors methodology v1.1 (RECIST). RECIST is developed for phase II trials as a surrogate endpoint for clinical benefit. However, this method has several drawbacks. It has a high inter-observer variation [1] and association with parameters of clinical benefit, overall survival or quality of life, is limited. Furthermore, RECIST lacks sensitivity and does not provide insight into the biological characteristics of the tumor, even though these characteristics gained increasing interest for treatment selection due to development of targeted therapies in the last decade. Circulating cell-free DNA (cfDNA) has great potential to overcome these drawbacks. cfDNA is present in the bloodstream and originates from both tumor cells and healthy cells. Unlike traditional tissue biopsies, liquid biopsies containing cfDNA can be obtained in a minimally invasive manner from for example peripheral blood, urine, cerebrospinal fluid or sputum and are therefore suited for longitudinal sampling. Circulating tumor DNA (ctDNA) reflects the genetic and epigenetic characteristics of a tumor and may therefore be used to predict response to treatment and development of treatment resistance. Interestingly, the proliferation rate of a tumor is associated with the amount of ctDNA [2], suggesting ctDNA levels can be used to monitor treatment response. Next to tumor-derived DNA fragments, the blood of patients with metastatic cancer can also contain circulating tumor cells (CTCs). CTCs can be detected using the FDA-approved CellSearch method. Cristofanilli et al. have shown the prognostic value of these CTCs in patients with MBC and found that two or more CTCs were detectable in 64 percent of MBC patients who started with second or subsequent line of therapy [3]. Compared to CTCs however, ctDNA is detectable in higher fractions of MBC patients and only requires standard laboratory equipment facilitating its use in daily clinical practice [4].

Different types of biomarkers can be used to detect ctDNA within the total pool of cfDNA, including tumor-specific somatic mutations [5], copy number variations (CNVs) [6], and methylation patterns [7], as well as cfDNA and concentrations. Regardless of the marker of choice, recent technical advances resulted in a wide variety of PCR-based and NGS-based methods for the detection of ctDNA, ranging from single markers and dedicated panels of markers to genome-wide approaches. Both digital PCR (dPCR) and quantitative PCR have a very low limit of detection, are cost-effective and require low hands-on time, making them well suited for large sample series. A drawback of these assays is that they can only detect a limited number of predefined alterations. Targeted or genome-wide NGS-sequencing requires more hands-on time, comes at a (substantially) higher cost, and generally requires more cfDNA. However, depending on sufficient sequencing depth, NGS can detect every alteration present within the included amplicons. The use of unique identifiers or the separation of amplicons into droplets, prior to amplification allows for the removal of sequencing artifacts introduced by PCR and thus increases analytical sensitivity particularly for somatic mutations.

Notwithstanding the recognition of the potential clinical value of cfDNA, so far cfDNA is not routinely used in clinical practice for MBC patients. To implement a test into daily clinical practice, a test must have shown analytical and clinical validity, and clinical utility. Analytical validity describes whether a test accurately detects the presence or absence of a certain feature (e.g., mutation, CNV, methylation). Clinical validity refers to the accuracy of a test in confirming the presence or absence of a particular clinical condition. Clinical utility is the added value of the test for diagnosing or clinical decision making. Currently, analytical validity has been demonstrated for a variety of cfDNA assays [8,9]. Therefore, this systematic review evaluates literature on clinical validity of cfDNA for monitoring treatment response in MBC patients during systemic treatment in peripheral blood. Furthermore, it will give a future perspective on the clinical utility of the potential applications of cfDNA in the metastatic setting.

## 2. Materials and Methods

### 2.1. Literature Search Strategy

This systematic review is reported according to PRISMA (Preferred Reporting Items for Systematic Reviews and Meta-Analysis) guidelines [10]. A literature search was performed on 15 October 2020 in the following databases: Embase, Medline, Web of Science, Cochrane and Google Scholar from 30 April 2010 onward. The search was developed with an experienced librarian. A combination of the following key words was used in the search: “cell free DNA”, “circulating tumor DNA”, “breast cancer”, “metastasized”, “monitoring”, “tumor load”. The search strategy was modified for every database (Appendix A).

### 2.2. Selection of Studies

Both prospective and retrospective studies in which monitoring was part of the study were included. Inclusion criteria were as follows: (1) the study reports on patients with metastasized or locally advanced breast cancer and response to therapy was measured by cfDNA on more than a single time point including a baseline sample, (2) the study had to be published in the last 10 years and, (3) had to be available in full text. Case reports or case series including only 1 or 2 patients were excluded.

Based on these criteria articles were screened and selected by two independent reviewers (E.J. and T.D.). In case of disagreement a third reviewer (S.W.) was consulted to reach consensus. 

### 2.3. Data Extraction

Data of the included studies was independently extracted by E.J. and T.D. by using a predefined extraction form. Information was extracted on: study population, number of patients, type of cfDNA assay(s) used, the measure of treatment response which was used (for example imaging or a tumor marker like CA-15.3), and information on clinical outcome. Any disagreement between the reviewers was resolved by discussion and led to consensus. 

### 2.4. Methodological Quality

After selection of the studies, the risk of bias was assessed by the QUIPS tool which incorporates selection bias, performance bias, detection bias, attrition bias, and reporting bias [11]. Bias based on attrition was omitted in the assessment since many articles evaluated cfDNA in retrospectively selected blood samples. To assess the overall risk of bias a total score based on the individual domains described above was determined. The method of this systematic review was added to PROSPERO (www.crd.york.ac.uk/PROSPERO (accessed on 15 March 2021)) in May 2020 (ID CRD42020185710 was assigned). 

## 3. Results

### 3.1. Literature Search and Selection

A total of 1750 publications were found in our initial search. After title and abstract screening, 172 publications were eligible for full text screening. Hereof, a total of 33 publications met our predetermined criteria and were qualified for inclusion in this review. The flowchart of our systematic review for study assessment is shown in Figure 1. Data extracted from the studies are shown in Table 1.

### 3.2. Characteristics of the Included Studies

The included studies varied in study design from phase 1 trials, which explored cfDNA to monitor the treatment response, to technical papers, which described the development and first validation of a specific cfDNA assay by comparing it with treatment response. Generally, two types of studies focusing on treatment response were included. Almost half of the included studies (15/33) related ctDNA dynamics between baseline and follow-up to progression free survival (PFS) (indicated as “endpoint” in Table 1), whereas the other studies (14/33) related real-time ctDNA data either with treatment response on CT or with a protein tumor marker like CA 15-3 (indicated as “continuous” in Table 1). The remaining four studies included both types of treatment response measurements. The included studies monitored different numbers of MBC patients (range: 4 to 129), during different lines and different types of treatment. Thirty studies monitored breast cancer patients exclusively, whereas the remaining three studies also included patients with other types of advanced cancer. From these studies, only data regarding the breast cancer patients were included in Table 1. 

Different types of blood collection tubes were used in the included studies (Figure 2). Remarkably, nine studies did not mention the type of tubes used, even though it is well known that the use of tubes without a cell stabilizing compound could increase leakage of genomic DNA from leukocytes into the plasma thereby influencing the variant allele frequency (VAF) of detected somatic mutations in plasma [12]. Isolation of cfDNA in the studies was predominantly performed by the QIAamp Circulating Nucleic Acid kit (Figure 3; 16 out of 33 studies). Overall, the variation in used cfDNA isolation methods was considerable, with 11 studies using a unique cfDNA isolation method not used in any other included study.

Different types of biomarkers originating from cfDNA were used for monitoring in the 33 included articles (Table 1). Only three studies monitored multiple types of biomarkers [13,14,15]. As described in the methods section, all studies which monitored treatment response longitudinally were included in this review, even though the studies differed in design and follow up. In some studies (14 out of 33), only patients with detected alterations in the cfDNA at baseline were evaluated longitudinally, while in the remaining studies (19 out of 33) all patients regardless of their baseline result were followed longitudinally. This difference in design influenced the fraction of patients in which the explored cfDNA assay could be applied and for this reason both a baseline and longitudinal column was added to Table 1 to indicate the number of patients in which the assays were explored and positive for the specific alteration.

**Table 1 cancers-13-01811-t001:** Published studies on cfDNA monitoring in MBC.

Detection	First Author	Year	Method	Target(s)	Baseline	%	Longitudinal	%	Correlative	Measurement	Determined by	Risk of Bias (QUIPS)
Mutation	J.A. Garcia-Saenz [16]	2017	dPCR	*PIK3CA*	8/32	25.0%	8/8 *	100.0%	Partial	Continuous	RECIST, PTM	Low
	W. Jacot [17]	2019	dPCR	*PIK3CA*	10/36	27.8%	10/36	27.8%	Yes	Endpoint	RECIST	Low
	A.R. Kodahl [18]	2018	dPCR	*PIK3CA*	20/60	33.3%	4/6 *	66.7%	Yes	Continuous	RECIST	Moderate
	X. Li [19]	2020	NGS	*ESR1*	9/45	20.0%	5/5 *	100.0%	Partial	Continuous	RECIST, PTM	Moderate
	P. Wang [20]	2015	dPCR	*ESR1*	7/29	24.1%	4/4 *	100.0%	Yes	Continuous	PTM	Moderate
	S.R. Vitale [21]	2018	dPCR	*ESR1*	3/67	4.5%	4/17	23.5%	Yes	Continuous	Not defined	Moderate
	C. Paoletti [22]	2018	dPCR	*ESR1*	14/45	31.1%	17/45	37.8%	Yes	Endpoint	RECIST, PTM	Low
	D. Sefrioui [23]	2015	dPCR	*ESR1*	4/7	57.1%	4/7	57.1%	Yes	Continuous	Not defined	Low
	E. Jeannot [24]	2020	dPCR	*ESR1*	17/59	28.8%	15/15 *	100.0%	Yes	Endpoint	RECIST	Low
	F. Clatot ^+^ [15]	2020	dPCR	*ESR1*	22/70	31.4%	22/22 *	100.0%	Yes	Continuous	RECIST	Low
	T. Takeshita [25]	2016	dPCR	*ESR1*	12/42	28.6%	12/42	28.6%	Yes	Both	RECIST, CA 15-3, CEA	Low
	T. Takeshita [26]	2017	dPCR	*PIK3CA*	17/69	24.6%	21/52	40.4%	No	Both	RECIST	Low
			““	*ESR1*	20/69	29.0%	24/52	46.2%	Yes	Both	““	
	J.M. Spoerke [27]	2016	dPCR	*PIK3CA*	62/156	39.7%	41/60	68.3%	Yes	Both	RECIST	Low
			““	*ESR1*	57/153	37.3%	42/60	70.0%	No	Both	““	
	B. O’Leary [5]	2018	dPCR	*PIK3CA*	100/455	22.0%	65/65 *	100.0%	Yes	Endpoint	RECIST	Low
			““	*ESR1*	114/445	25.6%	73/73 *	100.0%	No	““	““	
	J.S. Frenel [28]	2015	dPCR	*PIK3CA*	1/7	14.3%	1/7	14.3%	No	Endpoint	RECIST	Low
			““	*TP53*	5/7	71.4%	2/7	28.6%	Yes	““	““	
	S.J. Dawson [4]	2013	TAm-Seq	*PIK3CA* and/or *TP53*	24/52	46.2%	25/30	83.3%	Yes	Continuous	RECIST	Low
			““	*PIK3CA*	9/30	30.0%	11/30	36.7%	n.d.			
			““	*TP53*	15/30	50.0%	17/30	56.7%	n.d.			
	S. Hrebien [29]	2019	dPCR	*PIK3CA, GATA3, ESR1* and/or *TP53*	38/58	78.1%	35/35 *	100.0%	Yes	Endpoint	n.a.	Low
				*PIK3CA*	30/58	51.7%	32/35	91.4%	n.d.			
				*TP53*	4/58	6.9%	4/31	12.9%	n.d.			
	F. Ma ^+^ [14]	2016	NGS	*PIK3CA, mTOR, PTEN* and/or *TP53*	9/18	50.0%	11/18	61.1%	Yes	Endpoint	RECIST	Low
			““	*PIK3CA*	6/18	33.3%	8/18	44.4%	n.d.			
			““	*TP53*	3/18	16.7%	7/18	39.9%	n.d.			
	F. Ma [13]	2019	NGS	193 gene panel	37/37	100.0%	21/21	100.0%	Yes	Endpoint	RECIST	Low
			““	*PIK3CA*	13/37	35.1%	10/21	47.6%	n.d.			
			““	*TP53*	20/37	54.1%	13/21	61.9%	n.d.			
	S.W. Lok [30]	2018	dPCR	*PIK3CA, ESR1, GATA3* and/or *MAP3K1*	28/33	84.8%	28/33	84.8%	n.d.			Low
			““	*PIK3CA*	14/33	42.4%	14/33	42.4%	n.d.			
			““	*ESR1*	10/33	30.3%	10/33	30.3%	Yes	Both	RECIST	
			““	*GATA3*	5/33	15.2%	5/33	15.2%	n.d.			
			““	*MAP3K1*	4/33	12.1%	4/33	12.1%	n.d.			
	C.X. Ma [31]	2017	NGS	*HER2*	9/381	2.4%	11/11 *	100.0%	Yes	Continuous	RECIST	Low
			““	*PIK3CA*	n.a.		5/11	45.5%	n.d.			
			““	*ESR1*	n.a.		2/11	18.1%	n.d.			
			““	*TP53*	n.a.		6/11	54.5%	n.d.			
	C. Hufnagl [32]	2020	TAm-Seq	8 gene panel	4/4	100.0%	4/4	100.0%	No	Continuous	RECIST, CA 15-3	Moderate
	K. Page [33]	2016	dPCR	16 gene panel	21/42	50.0%	9/9 *	100.0%	Yes	Continuous	RECIST, CA 15-3	Moderate
			““	*PIK3CA*	12/42	28.6%	6/9	66.7%	n.d.			
			““	*ESR1*	6/42	14.3%	2/9	22.2%	n.d.			
			““	*TP53*	6/42	14.3%	2/9	22.2%	n.d.			
	R.D. Baird [34]	2019	TAm-Seq	20-gene panel	12/30	40%	4/4 *	100%	Yes	Continuous	RECIST	Low
			““	*PIK3CA*	7/30	23.3%	2/4	50%	n.d.			
			““	*ESR1*	3/30	10%	1/4	25%	n.d.			
			““	*TP53*	5/30	16.7%	2/4	50%	n.d.			
CNV	Guan [35]	2020	NGS	*HER2*	47/105	44.8%	19/26	73.1%	Yes	Endpoint	Not defined	Low
	B.S. Sorensen [36]	2010	qPCR	*HER2*	14/28	50.0%	22/22	100.0%	Yes	Endpoint	Not defined	Low
	F. Ma ^+^ [14]	2016	NGS	*HER2*	13/17	76.5%	13/17	76.5%	Yes	Continuous	RECIST	Low
	C. Suppan [6]	2019	FastSeq	*Line1*	10/29	34.5%	29/29	100.0%	Yes	Continuous	CA 15-3	Moderate
Methylation	M.J. Fackler [7]	2014	qMSP	Cumulative gene index (10 genes)	52/57	91.2%	13/13 *	100.0%	Yes	Continuous	RECIST	Low
							29/29 *	100.0%	Yes	Endpoint		
	K. Visvanathan [37]	2016	qMSP	Cumulative gene index (6/10 genes)	129/129	100.0%	129/129	100.0%	Yes	Endpoint	RECIST	Low
	M. Zurita [38]	2010	qMSP	*14-3-3-σ*	34/34	100.0%	34/34	100.0%	Yes	Endpoint	RECIST	Low
	S. Kristiansen [39]	2015	qMSP	*RASSF1A*	n.a.	n.a.	29/29	100.0%	Yes	Continuous	PTM	Moderate
			qMSP	*LINE-1*	n.a.	n.a.	n.a	n.a	No	Continuous	PTM	
	X.L. Liu [40]	2020	WGBS	Whole genome	16/16	100.0%	16/16	100.0%	Yes	Endpoint	Not defined	Moderate
*Size/Conc.*	Z. Ye [41]	2019	qPCR	*Alu115* and *Alu81*	117/117	100.0%	22/22	100.0%	Yes	Endpoint	RECIST	Low
	F. Clatot ^+^ [15]	2020	dPCR	cfDNA conc.	103/103	100.0%	70/70	100.0%	No	Continuous	RECIST	Low

The column “Baseline” shows the number of patients in which the described aberration is present (nominator) relative to the group of patients in which the aberration is determined at baseline (denominator). The column “Longitudinal” shows the number of patients in which the described aberration is present at any time point (nominator) relative to the group of patients in which the aberration is measured longitudinally (denominator). * These studies monitored only baseline-positive samples longitudinally. ^+^ These studies monitored two types of biomarkers Abbreviations list: dPCR = digital PCR, NGS = next generation sequencing, Tam-Seq = tagged amplicon deep sequencing, FastSeq = fast aneuploidy screening test-sequencing system, qMSP = quantitative methylation specific PCR, WGBS = whole genome bisulfite sequencing, RECIST = response evaluation criteria in solid tumors, PTM = protein tumor marker, n.a. = not applicable, n.d. = not determined.

### 3.3. Risk of Bias

The risk of bias was assessed in all 33 included studies, 24 studies had a low risk of bias, nine studies had moderate risk of bias and no studies had a high risk of bias according to the summarized QUIPS tool (Table 1).

### 3.4. Mutation-Based ctDNA Detection

Detection of tumor-specific mutations in cfDNA was most commonly used for monitoring in MBC patients (24 out of 33 studies), using two different techniques with high analytical sensitivity; digital PCR (dPCR; 17 out of 24 studies) and next generation sequencing (NGS; 7 out of 24 studies).

The somatic mutations which were assessed to evaluate response differ per study. Mutations in *PIK3CA*, *ESR1* and *TP53* were most frequently assessed (Table 1). In total, the presence of *PIK3CA*, *ESR1* and *TP53* mutations at baseline in the included studies ranged from 14.3–51.7%, 4.5–57.1%, and 6.9–54.1% respectively. This indicates that even in selected populations only around half of the patients could be monitored by use of mutations in a single gene. When a mutation in *PIK3CA* or *TP53* was present, dynamics in VAF or number of mutant molecules were associated with PFS or real-time response (CT-scan or protein tumor marker) in all included studies. The association with treatment response was less evident for dynamics in *ESR1* mutations. Two of the 15 studies which analyzed *ESR1* mutations did not find a relationship between dynamics in *ESR1* and treatment response [5,27]. These contradictory results are likely due to the fact that *ESR1* mutations are known to drive resistance to aromatase inhibitors [42]. Therefore, these mutations predict unresponsiveness to aromatase inhibitors specifically and may not reflect treatment response in general. In support of this, both O’Leary et al. and Spoerke et al. monitored patients treated either with fulvestrant and CDK4/6 inhibitor or with fulvestrant alone and showed that *ESR1* mutations were subclonal and not associated with treatment response, whereas *PIK3CA* mutations in the same patients were [5,27]. 

The use of a panel of genes could broaden the applicability of a test since the mutational landscape in MBC is heterogeneous [43]. In total 9 studies used a combination of genes as a monitoring tool, ranging from a combination of mutations in two genes up to a panel of 193 genes. All studies observed an association between the dynamics in mutations and treatment response during monitoring. Ma et al. report that the 193-gene panel used resulted in a strong positive correlation between mutational tumor burden index (mTBI) and treatment response [13]. The mTBI was used as a reflection of the percentage of ctDNA detected in cfDNA and was calculated based on the mean VAFs of mutations in a mutation cluster with the highest cellular prevalence of ctDNA. In their study the mTBI proved superior to single gene mutations for assessing therapeutic response. 

In summary, the heterogeneity in mutational landscape hampers the use of a limited number of hotspot mutations in MBC, but longitudinal monitoring of hotspot mutations does associate with outcome in all 24 studies.

### 3.5. Copy Number Based ctDNA Detection

Next to mutations and DNA methylation, CNVs represent another hallmark of the cancer genome. These CNVs often affect tumor suppressor genes or oncogenes, altering their expression and thereby driving tumor progression. We found four studies which used CNVs to evaluate treatment response. Three of the four studies focused on *HER2* amplifications during HER2-targeted therapies in patients with a *HER2* amplified primary tumor [14,35,36]. In all these three studies the level of *HER2* amplification in the cfDNA was associated with treatment response. 

The study of Suppan et al. determined genome-wide copy number profiles at low resolution using the mFast-SeqS method [6]. This method only requires a minimal cfDNA input (as low as 0.5 ng) and uses *LINE-1* elements to estimate copy numbers per chromosome arm as well as a genome-wide aneuploidy score. Although, in this study, in more than half of the included patients aneuploidy scores were low for all analyzed time points despite progressive disease. The genome-wide cfDNA aneuploidy-score was associated with treatment response based on RECIST and CA 15-3 at multiple time points. Together, these results suggest potential validity of this technique to monitor treatment response, but also demonstrate that CNV-based assays have a relatively high limit of detection for ctDNA.

### 3.6. DNA Methylation-Based ctDNA Detection

In general, tumor cells show overall genome-wide decrease in DNA methylation with focal hypermethylation in promoter regions of tumor suppressor genes [44]. Cancer-specific DNA methylation alterations are generally considered an early event in tumorigenesis and therefore are expected to have a high penetrance in the tumor. The fact that this process involves a greater part of the tumor genome than point mutations, suggests that genome-wide evaluation of ctDNA methylation patterns could lead to an enhanced sensitivity. 

Longitudinal monitoring of ctDNA methylation is associated with treatment response in all five included studies in this review. All studies detected DNA methylation using bisulfite conversion of patient cfDNA. ctDNA methylation detection using qMSP was done in four out of five of the studies evaluating various targets which are differentially methylated in MBC patients compared to healthy subjects [7,37,38,39].

Two studies by the same research group investigated the same six genes [7,37]. Fackler et al. started with a panel of 10 markers and show that methylation patterns of these 10 genes between two time points in 29 patients were associated with treatment response on the first CT scan after the ctDNA measurements [7]. In patients with a response or stable disease a significant decrease in methylation of the 10 genes was observed, whereas this decrease was not present in patients with progressive disease on the first CT scan. Disease was monitored real-time at three or more time points in 13 patients with ctDNA and imaging, and reflected treatment response based on the RECIST during the course of disease in 10 patients. Subsequently a reduced panel of six genes in the study of Visvanathan et al. [37] was used to successfully monitor disease progression in 129 patients. Only the study of Liu et al. used whole genome bisulfite sequencing to analyze the complete methylome [40]. Here, they calculated the methylation ratio and density per 200 kb region and ultimately identified one region of 200 kb in chromosome 6 which could accurately stratify patients in terms of PFS. Notably, even though only five of the included studies used ctDNA methylation markers, the level of methylation was associated with disease progression in all studies.

### 3.7. cfDNA Abundance

Two studies analyzed total cfDNA irrespective of measuring a cancer-specific biomarker to monitor MBC patients [15,41]. Ye et al. monitored *Alu* DNA elements, which was used to calculate the total cfDNA concentration [41]. The cfDNA concentration, irrespective of the cells of origin, was associated with disease outcome. The study of Clatot et al. investigated the correlation between *ESR1* mutation detection in cfDNA, CA 15-3 dynamics, or cfDNA dynamics and PFS [15]. They showed that, unlike *ESR1* mutational status and CA 15-3 dynamics, cfDNA concentration did not correlate significantly with disease outcome.

## 4. Discussion

This study is the first systematic review regarding the monitoring of treatment response in MBC patients in a longitudinal fashion using cfDNA. In general, the ctDNA level that was measured using different assays associated well with the treatment response measured by imaging or protein tumor markers. Studies focusing on the predictive role of early dynamics in ctDNA levels relative to a baseline sample found that a rapid decrease in ctDNA level was predictive for a favorable PFS.

### 4.1. Analytical Validity

Monitoring of treatment requires standardized procedures for collection and processing of baseline and follow-up samples at least within a patient but preferably also between patients and studies. The latter is particularly adamant to allow adequate comparison between studies necessary to expedite clinical implementation of ctDNA. The studies included here used different types of tubes and different types of cfDNA extraction methods. These two variables could potentially lead to variable fractions of ctDNA in the isolated cfDNA. Most of the studies used cell stabilizing tubes: tubes that stabilize nucleated cells thereby preventing the release of genomic DNA in the plasma. This is essential to compare fractions of ctDNA at multiple time points. The use of non-stabilizing tubes could, depending on the time of sample processing, lead to a higher fraction of genomic DNA in the blood plasma impacting the analysis [12]. With respect to cfDNA extraction kits, a recent multicenter comparison of cfDNA work flows found that the highest recovery was seen in the QIAamp Circulating Nucleic Acid kit, which is considered the current gold standard approach for cfDNA extraction. However semi-automated extraction protocols, barely used in the included studies, were found to perform most consistently in extracting cfDNA [45], which is crucial for longitudinal treatment response monitoring.

Additional necessary considerations with regard to especially the use of mutations for monitoring treatment response, include the unit of measurement and the determination of the origin of the mutated DNA. The included studies used different units of measurement for the ctDNA fraction, either VAF or the absolute number of mutant molecules per defined unit. The study of Bos et al. demonstrated that both units of measurement are heavily impacted by (pre-)analytical factors, which could limit their value for longitudinal monitoring [46]. Consequently, standardization of (pre-)analytical factors is required for successful future clinical implementation of ctDNA for longitudinal monitoring of treatment response.

A second important consideration is the origin of somatic mutations which are detected in cfDNA, since clonal hematopoiesis is a contributor to plasma DNA variants as previously described by Razavi et al. [47]. Analyzing matched white blood cells of patients could solve this issue but increases costs.

### 4.2. Clinical Validity

The studies included here differed with respect to both the type of alteration measured and the methods used to measure them. Each of the used approaches have their own strengths and limitations, which we summarized in Table 2. Furthermore, most studies included small numbers of patients and studies validating findings are currently lacking. However, detection of tumor-specific biomarkers greatly increases specificity compared to cfDNA levels, as is illustrated by the absence of a robust correlation between cfDNA levels and tumor load in the included studies. Mutation-based assays to monitor treatment response in patients with MBC are reported most often in the included studies. All studies found a relationship between treatment response measured by ctDNA and treatment response based on imaging. However, not all individual mutations associated equally well with treatment response, which is likely due to the subclonal presence of part of the mutations and the effect of prior and current treatments on the mutational landscape. Besides the polyclonal presence of mutations, an important drawback of mutation-based monitoring observed in the included studies was the limited number of patients with a mutation at baseline which could be traced. A partial solution for the genetic heterogeneity of advanced breast tumors in monitoring treatment response by using ctDNA could be the use of targeted panels rather than single mutations. Although broad panels of genes are required to screen the heterogeneous landscape of advanced breast cancer [43]. 

Based on the few included studies investigating DNA methylation, cfDNA methylation appears a promising method for monitoring disease in MBC. Up to data most of these studies focused on only a few methylation markers, whereas most profit in sensitivity is expected from genome-wide methylation patterns. Indeed, results from the Circulating Cell-free Genome Atlas (CCGA; NCT02889978) study indicate that cfDNA methylation analyses enable more sensitive detection of early stage cancers compared to CNVs and mutations, although breast cancer was not included in this study [40]. 

This supports the potential of using methylation patterns to monitor treatment response in both advanced and early breast cancer. The limited number of studies focusing on methylation and the high variance in biomarker selection in these studies demonstrates that cfDNA methylation is still in its infancy and no universal cfDNA methylation biomarkers for breast cancer have emerged yet.

Finally, a limited number of studies focused on CNVs to measure treatment response in the advanced setting. These studies focused mainly on monitoring of HER2-targeted treatment in HER2-positive breast cancer patients by detection of *HER2* copy numbers in cfDNA. In these studies, the presence of HER2 is a predictor for response to anti-HER2 therapy and the quantification of the level of *HER2* amplification could be used as a marker of response to therapy. This method of monitoring is consequently limited to breast cancer patients with HER2+ tumors. Only one of the included studies focused on genome-wide chromosomal aneuploidy and its results are therefore applicable in patients with all types of breast cancer [6]. Unfortunately, sensitivity of this assay appears limited, which is in concordance with previous studies demonstrating that ctDNA detection based on CNVs requires a ctDNA fraction of at least 5–10 percent [48,49]. This drawback makes the assay less suitable for patients with low ctDNA levels and advocates limiting its application to advanced stage patients.

### 4.3. Clinical Utility

Several ongoing and recently published studies focus on the clinical utility of ctDNA in finding and targeting specific alterations or monitoring and treating resistance mutations (Table 3). For this purpose, especially monitoring of targetable mutations or CNVs is worthwhile. Mutations in *PIK3CA* are an example, since alpelisib (PIK3CA inhibitor) has shown to be effective only in HR+/HER2− MBC patients carrying a mutation in *PIK3CA* [50]. Retrospective analyses in this study showed that patients with a *PIK3CA* mutation detected in ctDNA benefit more from addition of alpelisib compared to all patients in which *PIK3CA* mutations were detected in tumor tissue (primary tumor or metastasis) [51]. 

To the best of our knowledge, no studies focusing on clinical utility of ctDNA as a measure of treatment response are published or currently ongoing. This is remarkable since ctDNA assays could potentially also have important clinical implications in this area. ctDNA assays could potentially play a role in specific clinical settings, for example during the first lines of treatment of patients with HR+HER2− MBC, in which the PFS of a single treatment line could be a few years. Monitoring of treatment by a ctDNA assay could complement or potentially even partially replace the three-monthly performed CT-scans. This will require an assay with a sensitivity approximating 100 percent and in case of progression additional imaging will still be indicated to assess for example fracture risk. The implementation of ctDNA assays in this setting could improve the timely detection of progression since ctDNA assays are in general more sensitive than imaging [54]. In addition, it may reduce the number of hospital visits and health care costs for these patients. Further research is required to explore whether the use of ctDNA assays for earlier detection of progression will actually lead to a benefit for patients in terms of quality of life and/or survival. A previous study showed that early switching to an alternative cytotoxic chemotherapy based on persistently increased numbers of circulating tumor cells in their blood did not improve overall survival in these patients [55].

Another clinical application of monitoring tumor load by using ctDNA is to use dynamics in ctDNA in the first weeks after initiating systemic treatment for early response monitoring. This could reduce anxiety in patients and limit unnecessary toxicity. Studies focusing on this approach for guiding therapy are not yet performed or ongoing in MBC to our knowledge.

### 4.4. Limitations

This review has several limitations. One of these limitations is the rather limited number of patients and patient samples included in the mentioned studies. Furthermore, the heterogeneity in study design prohibits a formal meta-analysis. Several studies focused on assay development for ctDNA detection and validated their developed assay in a small cohort of retrospective patient samples, which may lead to bias. Moreover a few studies used only a tumor marker to assess treatment response and did not use any imaging for disease evaluation or RECIST, the current gold standard to determine PFS.

## 5. Conclusions

This systematic review showed an association between ctDNA and treatment response in most of the included studies although a majority of the included studies only included low numbers of patients. To enable implementation of ctDNA for tumor load monitoring in daily clinical practice for MBC patients, studies focusing on standardizing operating procedures, uniform approaches of detecting ctDNA, and prospectively exploring the role of ctDNA in large numbers of patients with well-documented clinical follow up are urgently needed, as suggested previously [56]. This review further substantiated this need by systematically evaluating the existing literature on monitoring disease response by ctDNA. It is encouraging that an increasing number of phase 1, 2, and 3 studies incorporate the collection of plasma during treatment and follow-up for cfDNA analysis. Ultimately, this will lead to more insight in clinical validity and utility of cfDNA. 

Considering all the evidence provided in this review with respect to the detection of somatic mutations, methylation markers, and CNVs, clinical validity is currently only reached for assays detecting somatic mutations. However, based on the limited amount of data currently available, detection of cancer-specific methylation patterns in cfDNA appears particularly promising for treatment response monitoring and may prove to be superior with respect to sensitivity and applicability. For future ctDNA applications involving detection of minimal residual disease in patients in the adjuvant setting, a high sensitivity and applicability is also required. However, on the other hand, detection of targetable alterations like specific mutations could guide treatment choices in these patients to those with the potential to cure [54].

## Figures and Tables

**Figure 1 cancers-13-01811-f001:**
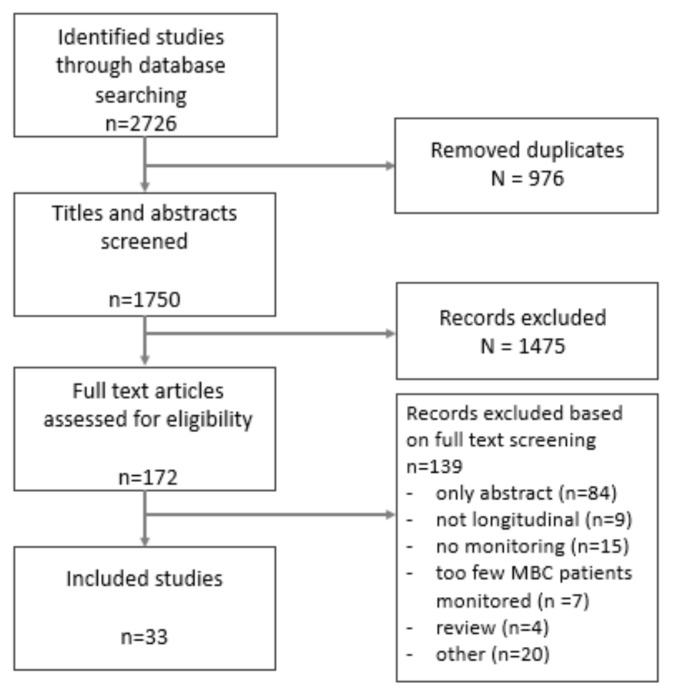
Flowchart of selection of studies.

**Figure 2 cancers-13-01811-f002:**
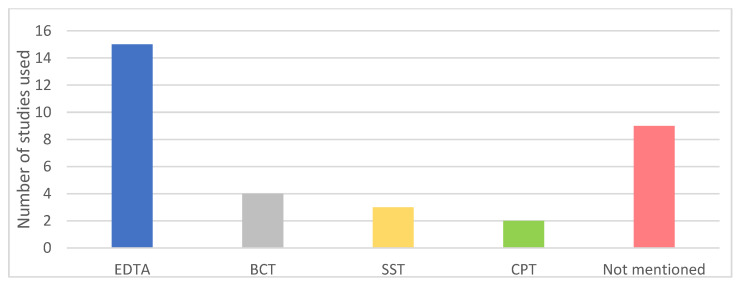
Types of collection tubes used in included studies. EDTA: Ethylenediamine tetraacetic acid; SST: serum-separating tube; CPT: cell preparation tube; BCT: blood collection tube with a preservative stabilizing nucleated blood cells (Streck).

**Figure 3 cancers-13-01811-f003:**
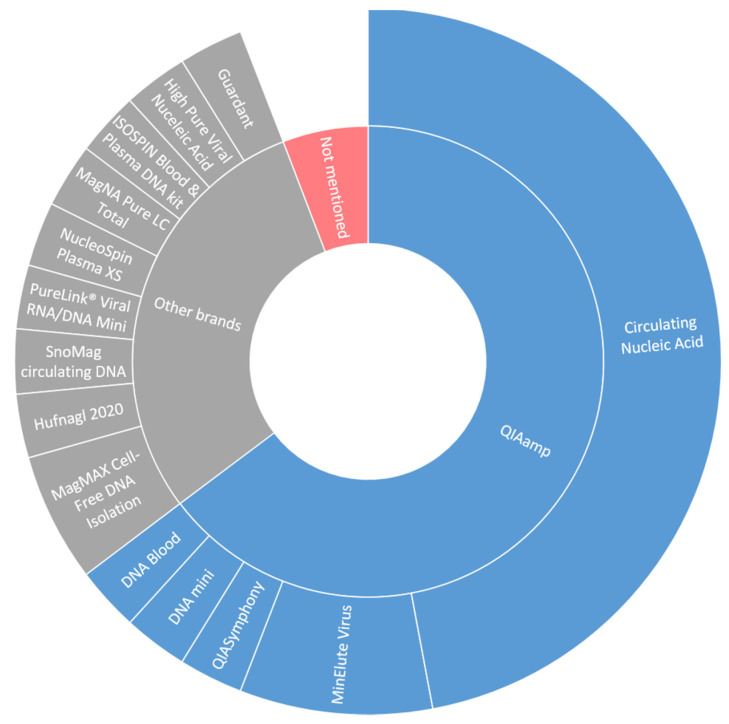
DNA isolation techniques in included studies.

**Table 2 cancers-13-01811-t002:** Advantages and drawbacks of different cfDNA assays.

Assay	Advantages	Drawbacks
Mutations	Cancer-specific Correlates well with treatment response Can detect (well-documented) clinically actionable mutations Can detect/identify therapy-resistant mutations	Subclonal mutations might not reflect tumor response accurately Mutational prevalence low for certain genes Prior knowledge on mutation-status needed for targeted approaches Need for germline analysis to exclude identification of non-cancer related mutations from clonal haematopoiesis
DNA Methylation	Disease-specific Correlates well with tumor response Robust alteration with high penetrance over course of disease Also measurable in early disease	Challenging on small amounts of cfDNA due to indirect detection
Copy number variations	Disease-specific Correlates well with tumor response Measurable in all patients of their respective cohorts	ctDNA fraction of 5–10 percent required for adequate detection
cfDNA concentration	Universally applicable to all patients Not limited to dedicated biomarkers/assays	Not disease-specific Limited correlation with tumor response Only applicable for monitoring

**Table 3 cancers-13-01811-t003:** Ongoing studies focusing on clinical utility of ctDNA in metastatic breast cancer.

ctDNA Application	Study Name/Author	Status	Patients	Intervention	Control	Outcome
Acting on resistance mutations	PADA-1 (NCT03079011)	Ongoing	800	Palbociclib treated patient with rising *ESR1* mutation levels will receive additional fulvestrant	Patients will continue with palbociclib. A subset will be crossed over	t.b.a.
	INTERACT Study (NCT04256941)	Ongoing	124	AI and CDK4/6i treated patients with *ESR1* mutation after 12 months will switch to fulvestrant	Patients will continue with AI	t.b.a.
Targeting actionable mutations	PlasmaMATCH [52]	Published	1150	Patients with detected mutations will enter a specific treatment cohort: (1) *ESR1*, fulvestrant; (2) HER2, neratinib; (3) AKT (and ER+), capivasertib plus fulvestrant; (4) AKT pathway activation, capivasertib monotherapy	None	357 patients (34%) with targetable mutation and 136 patients (13%) included in a treatment cohort
	Zivanovic et al. [53]	Published	234	Treatment based on detected actionable mutations	None	104 patients (44%) with actionable mutations, clinical management was changed in 40 patients (17%)

Search: August 2020 “metastatic breast cancer” “ctDNA. t.b.a. = To be announced.

## Data Availability

No new data were created or analyzed in this study. Data sharing is not applicable to this article.

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
