# Peer review of "A Systematic Review of the Use of Circulating Cell-Free DNA Dynamics to Monitor Response to Treatment in Metastatic Breast Cancer Patients"

_cancers, 2021, doi:10.3390/cancers13081811_

Round 1

Reviewer 1 Report

Dear authors, the current manuscript is well written, well organized and easy to read. Moreover the authors exposed clearly the limitations of the study.  However I would like to make some comments and ask for some clarifications from the authors.

  • the included studies have a big range of monitored MBC patients (4-455), this represents a huge difference in terms of significance. Moreover, the number of patients in each study could be included as information in table 1, since is quite relevant.
  • Why authors included 3 studies of other types of advance cancer?
  • CTCs, another liquid biopsy, is currently used in clinics for monitoring of breast cancer (using cell search approach). Why the authors did not mention this ? It would be interesting to make a point discussing the advantages of ctDNA over CTCs as liquid biopsy. 
  • The main conclusion of the authors is that ctDNA needs standarization of procedures and validation in a bigger number of patients. This has been already concluded by other recent studies such as the one by Tellez-Gabriel, M et al, IJMS.

Reviewer 2 Report

This systematic review by Jongbloed et al. investigated blood-derived ctDNA   and how they relate to disease response in metastatic breast cancer using standard tools in terms of systematic reviews. The review is executed well methodologically and nicely written. I would recommend this paper for acceptance without additional work.

Reviewer 3 Report

The manuscript by Jongbloed et al. presents a thorough systematic review in the field of ctDNA/cfDNA application for monitoring treatment and breast cancer progression. The collection of publications is well described, bias is assessed using QUIPS.  The authors mentioned limitation of their analysis and ctDNA/cfDNA based liquid biopsy approach. Important factors for future design of gene panels  are mentioned. I only have minor comments.

  1. The authors mention that there were different approaches in the inclusion of patients for longitudinal testing – sometimes all patients were tested at multiple time points, sometimes patients positive at baseline. Does it mean that all patients mentioned in Table 1, column “Longitudinal” were positive at baseline? Please add that information.

  1. Please indicate the total number of patients in which the analysis was performed in the fragment mentioned below.

“Two studies by the same research group investigated the same six genes [5, 36]. Fackler et al. started with a panel of 10 markers and show that methylation patterns of these 10 genes reflected both tumor burden based on the RECIST and reflect disease progression during the course of disease in 13 patients with metastasized breast cancer [5].”

  1. As the authors mention, clonal haematopoiesis might contribute to the frequency of detected mutated alleles in ctDNA, it is an important limitation that should be listed in Table 2 section “Mutations”.
